# Overdose Detection Technologies to Reduce Solitary Overdose Deaths: A Literature Review

**DOI:** 10.3390/ijerph20021230

**Published:** 2023-01-10

**Authors:** Alexa Rose Lombardi, Ritikraj Arya, Joseph G. Rosen, Erin Thompson, Ralph Welwean, Jessica Tardif, Josiah D. Rich, Ju Nyeong Park

**Affiliations:** 1Department of Biology, Boston College, Chestnut Hill, MA 02467, USA; 2Division of Biology and Medicine, Brown University, Providence, RI 02912, USA; 3Department of International Health, Bloomberg School of Public Health, Johns Hopkins University, Baltimore, MD 21218, USA; 4Harm Reduction Innovation Lab, Rhode Island Hospital, Providence, RI 02903, USA; 5School of Public Health, Brown University, Providence, RI 02912, USA; 6Center of Biomedical Research Excellence on Opioids and Overdose, Rhode Island Hospital, Providence, RI 02903, USA; 7Warren Alpert Medical School, Brown University, Providence, RI 02912, USA

**Keywords:** fatal overdose, room-based devices, mobile applications, wearable electronic devices, overdose detection technology

## Abstract

Drug overdoses were a leading cause of injury and death in the United States in 2021. Solitary drug use and solitary overdose deaths have remained persistent challenges warranting additional attention throughout the overdose epidemic. The goal of this narrative review is to describe recent global innovations in overdose detection technologies (ODT) enabling rapid responses to overdose events, especially for people who use drugs alone. We found that only a small number of technologies designed to assist in overdose detection and response are currently commercially available, though several are in the early stages of development. Research, development, and scale-up of practical, cost-effective ODTs remains a public health imperative. Equipping places where people live, learn, work, worship, and play with the necessary tools to detect and prevent overdose deaths could complement ongoing overdose prevention efforts.

## 1. Introduction

Unintentional drug overdose is the leading cause of injury-related death in the United States (US) [1], with nearly 92,000 fatalities recorded in 2020 [2] and over 100,000 fatalities in 2021 [3]. Drug overdose, particularly those involving illicit fentanyl and stimulants, has emerged as a challenging health and safety concern in the U.S. [4], with many deaths occurring in the absence of bystander intervention.

During an opioid overdose, an individual’s breathing slows down and can stop completely. An opioid overdose, including fentanyl overdoses, can be reversed *if* identification and action are taken within minutes (i.e., with the administration of naloxone, rescue breaths, and/or the application of supplemental oxygen). Naloxone rapidly reverses an opioid overdose by attaching to opioid receptors, displacing and blocking the effects of opioid agonists [5]. While naloxone is highly effective at reversing opioid overdoses, it is not effective in treating overdoses caused by benzodiazepines or stimulants (e.g., cocaine and amphetamines) [6]. For example, xylazine is a non-opioid veterinary tranquilizer increasingly involved in overdose deaths across the US. People are exposed to xylazine (knowingly or unknowingly) frequently in combination with opioids, particularly fentanyl, which causes sedation that may not be reversable with naloxone [7]. In the absence of supervised consumption spaces [8], using technology to connect people who use drugs with trained lay responders who are pre-equipped with naloxone and instructions for informing emergency medical services (EMS) is a highly effective harm reduction strategy for reducing the risk of fatal overdose.

People who use drugs alone are limited in their access to bystander intervention in the event of an overdose. Supervised consumption spaces provide individuals with the space to use drugs under supervision but face multiple social and legal barriers in the US. With growing knowledge of the reversibility of opioid overdoses, and the importance of timely response by EMS in the case of non-opioid overdoses or other non-overdose medical emergencies, different forms of overdose detection technologies (ODT) have been developed and some have become commercially available in recent years.

In this narrative review, we summarize several types of ODTs in different stages of development. Our goal is to inform communities and organizations that have responded to overdoses or that are concerned about the potential occurrence of overdoses in specific locations. We grouped ODTs into three categories: fixed-location devices, smartphone applications and hotlines, and wearable technology. Although some of these technologies have existed and been used by communities without recognition or formal evaluation, empirical evidence is needed to determine the feasibility of scaling up these services. While each differs in its approach and considers different preferences, these ODTs may provide resources to improve safety during episodes of drug use, especially in the absence of other overdose interventions. 

## 2. Materials and Methods

Our search strategy, implemented in October 2022, included two major constructs: “drug overdose” and “detection technologies”. Keywords, derived from Medical Subject Heading (MeSH) terms, under the “drug overdose” construct were “drug overdose”, “opiate overdose”, and “overdose”. Keywords under the “detection technologies” construct included “biosensing techniques”, “virtual monitoring”, “mobile applications”, “smartphone”, “reverse motion detector”, and “wearable electronic devices”. We implemented the search in PubMed and included all retrieved English-language peer-reviewed medical and public health literature. Inclusion criteria were articles that reported on ODTs. Exclusion criteria included articles that focused on unrelated concepts such as diabetes and hepatology; moreover, articles that only presented preclinical proof-of-concept studies (e.g., animal studies) were excluded. We also corroborated the results with a web search on the ODTs identified, and the research team’s knowledge of existing technologies that did not have published peer-reviewed articles at the time of the search. Eligible articles were selected and included in the review, while ineligible articles were recorded but filtered out.

## 3. Results

The finalized search strategy yielded a total of 49 articles; 18 were included and 31 were excluded. 

Among the ODTs identified, we found that although each type of technology triggers an alert in different ways, ODTs consistently connect people who use drugs to staff, peer, or community support if they are non-responsive, usually within minutes. These various tools are tailored to the priorities and needs of people who use drugs wherever possible, such as maintaining anonymity and confidentiality, and respecting personal preferences regarding bystander response and other types of support provided. The newest developments in technology-based contingency management provide improved scalability and lower costs than prior contingency management approaches [9]. As discussed below, ODTs can be grouped into three categories: fixed-location devices requiring physical installation, mobile applications and hotlines, and wearable technology.

### 3.1. Fixed-Location Devices Requiring Physical Installation

Fixed-location devices are physical devices installed in a high-traffic location for substance use and overdose. The Brave Sensor [10], released in 2018, is a contactless device installed by organizations in enclosed spaces such as single-stall restrooms to automate restroom safety precautions. Radar sensors, such as the Brave Sensor, solely use motion to detect a person’s movement, and do not infringe on their right to privacy, a key factor in development [11]. The device uses ultra-sensitive radar sensors to monitor micro-movements, like breathing, from the time a person enters the space until they exit. The timer length is predetermined by the organization and is customized to begin counting down when motions are initially identified. The sensor promptly alerts a designated responder phone [12] when motion stops or someone has been in a restroom/isolated space for a prolonged period [10].

An electrician from Andover, Massachusetts, personally designed and installed anti-motion detectors in four high-traffic bathrooms in the South End Clinic for individuals experiencing homelessness [13]. This sensor functions similarly to the Brave Sensor, with a two-minute and fifty-second timer. Rather than alerting staff via text message, an alarm sounds with a bright light flashing above the door [14]. Public restrooms are a widespread source of overdoses in North America [11], and the Brave Sensor and South End Clinic Anti-Motion Detector facilitate rapid response to solitary overdoses in these spaces. These types of sensors, also called reverse motion detectors [15,16], enhance the ability of organizations to provide higher-quality overdose monitoring and, in the event of a potential overdose, automatically alert staff. Text-message alerts are easily integrated into the organization’s workplace environment without altering the staff’s day-to-day responsibilities [17], and the more abrupt alarm is acceptable among staff in the South End Clinic.

The Brave Button system is a non-invasive and discreet way for residents of supportive housing facilities to contact support in the event of a potential emergency, including drug overdoses. Brave Buttons were developed for a variety of settings including multi-unit supportive housing facilities and scattered site housing projects [18]. The system consists of three components: buttons, hubs, and responder phones. These three elements work in tandem to connect residents to support while maintaining personal privacy [18]. The buttons are small wireless devices anchored to walls at an accessible height and location within supportive housing units. Residents press the button to contact the designated response team [19]. Buttons have two possible options for different circumstances when pressed: regular (button is pressed *once*) and emergency (button is pressed *twice or more*) [20]. The regular setting indicates that the resident would like support within the next five minutes, while the emergency setting specifies that a response is needed urgently. 

When the button is pressed, a message is sent to hubs located throughout the facility. The number of hubs within each facility depends on its relative size. The hub picks up a signal coming from a button and notifies the responder’s phone via text message with the necessary details including the urgency of the response and its exact location [19]. The responder is responsible for replying to this message with “Ok” to indicate that they are responding and will receive a second message asking for details on the scenario to categorize their response. If the designated responder does not act quickly, the Fallback Phone, monitored by managers or supervisors, is activated [20]. Primarily developed to facilitate rapid responses to opioid-involved overdoses, this system could also be helpful in other emergency scenarios. It is clear that “immediate housing-based interventions are needed to lessen the negative consequences” [14], and the Brave Button not only alleviates solitary overdose deaths in this space but also other non-overdose emergencies that may occur in these spaces, such as violence.

Push-activated intercom systems stem from various other brands with slight differences, but all have one common goal: a direct line of communication between people who are using drugs and supporting staff. In medical settings, intercoms can be valuable to save critical seconds getting to a patient in need [21]. Similarly, these devices are applied to detect overdoses in the locations where they are installed. When pressed, the intercom immediately opens an audio and/or visual call (depending on the system) from either side to monitor the needs of the caller. Some systems integrate communication between rescue services and rooms through one system and include acoustic room monitoring and surveillance [21]. 

The Health Resources in Action, Inc. (HRiA) includes intercom systems in its short list of bathroom safety recommendations to prevent fatal opioid overdoses. They inform readers to install intercoms into public bathrooms to communicate with people using the bathroom without having to knock/enter to ensure their safety [22] and include that they are best practiced when complemented with timers or time limit policies such as staff monitoring every 3–5 min [22]. The Washington Heights CORNER Project (WHCP), a syringe services program (SSP) in New York, paved the way for intercom monitoring before these guidelines were even implemented. In bathrooms at this SSP, an intercom is installed, and every three minutes, a trained staff member checks in on the caller to ensure they are still conscious [23]. If a response is not heard from the caller, the staff will press a button to unlock the door and enter to check on the well-being of the participant and administer naloxone in the case of an overdose [23]. People who have used the WHCP bathroom for this purpose indicate that it keeps people who use drugs safe and secure because they know the staff will not let anything bad happen to them [24]. Additionally, in a study of public injection drug use in New York City, interviewees indicated that the safest places to inject were syringe services program bathrooms with intercom systems, because “they’re always checking you in the bathroom” [25]. Intercoms are an affordable and acceptable option for locations looking to implement bathroom monitoring. 

Fixed-location devices requiring *installation* are limited for this reason (Table 1). Installation is both financially costly and time-consuming, so they are not always the most practical option for non-profit organizations. Additionally, push-activated systems require people who use drugs to engage prior to using their drugs, which could happen many times per day. This extra step would require training and reinforcement. While intercom systems add a private form of communication between staff and people who use drugs, it requires trained staff designated to respond to the intercom every few minutes. Other ODTs, discussed below, offer alternative benefits, and avoid some of these limitations, but have their own challenges.

### 3.2. Mobile Applications and Hotlines

Mobile applications (“apps”) and hotlines are ODTs in which a physical device is not necessary to prevent solitary overdose deaths. These devices are free to download or simply a hotline which individuals can call. 

The Never Use Alone (NUA) hotline began in August 2019 with the goal of being available in all 50 states in the US. As the COVID-19 pandemic continued unfolding, individual states including New York, Massachusetts, and Vermont started their own hotlines in April 2020. In 2021, NUA New England was formed from the merging of NUA Massachusetts and NUA Vermont and has since expanded its coverage to most of New England, including Maine, Massachusetts, New Hampshire, Rhode Island, and Vermont [26]. The hotline is monitored 24/7 and offers added anonymous protection for people who use drugs alone to prevent solitary overdose deaths, also called a virtual supervised consumption site [27]. People who use drugs in any state can contact the hotlines for their services; however, regional hotlines are being advertised to the public. 

When someone is about to use drugs alone, they call the hotline and are greeted by a trained operator who stays on the phone until the person is no longer at risk of overdose. Once connected, the operator elicits personal information from the caller including their first name (real or fake), the phone number they are calling from, the exact location from where they are calling, and the substance(s) they believe they are using as well as the method(s) by which they are consuming them [28]. If the caller does not respond for 30 to 45 s, the operator is trained to contact EMS and provide them with their exact location. EMS is informed of an “unresponsive person” at the given location [29] as opposed to an overdose, which reduces the chances of law enforcement involvement and, oftentimes, results in a faster response from EMS [28]. While NUA does not claim to guarantee safety, over 4400 calls have been received since it was founded in 2019, with 28 EMS calls, and all 28 of those lives were saved [27]. 

The National Overdose Response Service (NORS) is the NUA equivalent in Canada [30]. Both hotlines are toll-free numbers, which do not require callers to have minutes on their devices in order to make phone calls. From 15 December 2020, to 28 February 2022, NORS operators monitored over 2100 substance use calls with 53 resulting in emergency response activation [30].

The Canary—Prevent Overdose app functions by monitoring the user’s inactivity after activation using an accelerometer within their smartphone [31]. The user places their phone in their pocket once activated. According to the developer, the device detects and tracks small body movements, and if movements consistent with respiration are undetectable, Canary warns the user [32]. If the user does not respond to the prompts from the app, Canary alerts others of a potential overdose and information to assist them [32]. The privacy settings within the app allow the user to tailor their potential overdose alerts to their comfort level, which include a loud audible alert to people in the area and/or a discreet message to trusted contacts [32]. The app can be reset by turning off the alarm or by moving or toggling it off manually [31]. The Canary app is a customizable experience for the end-user to connect them to resources without having to directly communicate with any predetermined supporters. Some people may prefer the convenience of the Canary app, as it does not rely on specific geographical locations or the availability of volunteers.

The Brave App, previously called BeSafe, is an app that functions similarly to a virtual supervised consumption space. The app was designed for people who use drugs by people with lived experience, with careful consideration placed on maintaining anonymity and culturally appropriate support [33]. When the app is downloaded to an Apple or Android device, someone can start a call with a trained volunteer who can remotely monitor them during their use of drugs on the call and activate an emergency response if deemed necessary [34]. The overdose steps are chosen by the caller before they begin to use drugs, and it is known as their “game plan.” The game plan honors the caller’s expertise, autonomy, and dignity, ensuring that they have control over when, how, and if their supporter sends help [33]. Some callers prefer EMS while others prefer a roommate or neighbor [34]. The operator checks in throughout the call, and if the caller becomes unresponsive within 20 s, they gain access to the game plan [34]. The Brave App is unique in that callers do not need to disclose *any* personal information to use the service, and the entire process can be kept private and anonymous [34], a major concern expressed regarding smartphone applications such as ODTs [35]. Since its implementation, the Brave App has contributed to reducing overdose deaths and is recommended as a form of harm reduction that saves lives [36].

UnityPhilly is a smartphone application developed by Drexel University which first piloted a study in Philadelphia between March 2019 and February 2020 [37]. The app is different from the previously presented technologies, as it organizes and mobilizes a virtually connected community of people equipped with naloxone and ready to respond [38]. Individuals anticipating an overdose or bystanders suspecting one can send an emergency signal with a simple swipe on the app. In addition to calling 911 and sending EMS to the scene, everyone with UnityPhilly in the area is sent an overdose alert with the location details and encouraged to respond with naloxone [39]. A study conducted on overdose reversal by UnityPhilly users demonstrated high support for equipping as many people as possible with naloxone [40]. Additionally, studies demonstrate the acceptability of this overdose emergency alert format from people who use opioids due to their sense of community [41]. In a year-long pilot study, naloxone administration by informed volunteers occurred more than 5 min prior to EMS arrival in 59% of instances [42]. By monitoring geographic trends within Philadelphia and prioritizing hotpots of overdose, the community can implement interventions tailored to the needs of the neighborhood [42]. The speed with which naloxone is administered, varying even by just a few minutes, is the difference between an overdose near-miss and a fatality, and this communal effort increases the likelihood that someone can respond in the event of a solitary overdose.

The Lifeguard App was developed for use in British Columbia to address rising overdose deaths in the region. The Lifeguard App has a similar approach to previously mentioned applications and can directly link people who use drugs to responders if necessary. Before using drugs, app users activate the app to begin monitoring. After 50 s, the Lifeguard App will sound an alarm which can be toggled off by the user [43]. If the user is unable to stop the alarm to demonstrate consciousness, the alarm becomes louder. After 75 s of unresponsiveness, the app sends a phone call to EMS dispatchers with a text-to-voice explanation of the individual’s location [12,43]. The details of location include both the address and the user’s precise location within that address [44]. The application also includes instructions on how to administer naloxone, if available, or how to perform CPR, if applicable, for a bystander available to respond [44]. 

The Second Chance App is a smartphone app that creatively utilizes smartphone functions to rapidly respond to potential opioid overdoses. This app was developed by the University of Washington, and it can detect someone’s breathing from up to three feet away [45]. The app sends inaudible sound waves from the phone to the person’s chest and monitors how these waves return to determine the person’s breathing pattern [46,47]. If abnormal, decreased, or absent breathing is detected, the app sends an alarm requesting the individual to interact, and if they are unresponsive, Second Chance contacts emergency services and/or a previously identified supporter to respond [48]. Creators indicated their purpose for choosing the smartphone application format because most individuals own a smartphone and can access harm reduction services from the convenience of their mobile devices. The inequities of smartphone access among people who use drugs, specifically for those who are unhoused, remain a challenge to widespread adoption for all ODT smartphone applications [35]. 

iKeepr is a digital health application with similar main functions to the previously discussed technologies, connecting people who use drugs to elicited emergency contacts for immediate medical response [49]. The app user begins by “Setting a Safety Reminder” which includes a timer. At the previously selected time of the alert, this reminder triggers an emergency alarm, and if no response is indicated in the app, emergency notifications are sent to Guardian Angels (GA) [49]. GAs are the emergency contacts that are preselected by the person using the application. The GAs are prompted to accept the emergency request. One GA heads towards the location of the emergency with a naloxone kit, with instructions for various forms of administration detailed in the application, while another calls EMS with details of the emergency [49]. iKeepr is yet another example of how smartphone applications can provide connections to emergency medical responses to people who use substances alone.

Smartphone applications and hotlines, while currently free to individuals and organizations, do have their own limitations. Individuals are required to have phone credit, as well as a charged phone, for hotlines, and a charged smartphone, for applications (Table 1). These limitations require some planning on the part of the individual such as charging their device, application download, and having the application readily available, which keeps them from being applicable to all scenarios. Additionally, response time by EMS can vary across geographic contexts and may not be the best option for all individuals such as those in rural areas. Hotlines and applications depend on the caller’s responsiveness and preset thresholds, so premature activation of EMS may occur, which could make some individuals hesitant to use the services, despite their benefits. For example, the Canary app relies on a friend or family member’s willingness to intervene or contact EMS, which is not practical for many people who use drugs experiencing stigma or facing exclusion from familial and peer networks.

### 3.3. Wearable Technology

Wearable technologies circumvent some of the aforementioned limitations but bring additional challenges. Though less utilized, wearable biosensors have been studied for reversing overdoses since 2016, though early issues around accuracy were raised [50,51]. Today, these ODTs are less obtrusive and can stream information to a smartphone including precise measurement of timing, location, context, and duration of drug use [51,52]. However, in comparison to smartphones, smartwatches, personal data assistants, and beepers, wearable ODTs were noted as potentially stigmatizing electronic devices if outwardly visible [52].

To camouflage the apparatus, Embrace by Empatica developed a device similar in size and shape to a wristwatch that monitors physiological factors such as electrodermal activity (EDA), skin temperature, and locomotion [51]. This sensor is approximately 3 × 4 cm and is securely fastened to the wearer’s wrist [53]. Embrace sends information to the user’s smartphone to accurately indicate the user’s moments of greatest need [52]. Furthermore, this device provides two applications. The first app alerts the user of changes in autonomic stress levels, while the second alerts the user and/or nominated emergency contact of combined motion-autonomic events [53]. The HopeBand and iHeal are functionally similar to Embrace by tracking trigger points for opioid use, relapse, or overdose directly on the individual [54].

Wearable naloxone auto-injectors detect physiological factors to determine if and when naloxone is necessary. However, this device delivers a large dose of naloxone beneath the skin upon the detection of apnea in an individual [55] rather than simply alerting the individual to changes in their vital signs. The burst-release design has a clear advantage due to its automatic administration of naloxone [56] in response to opioid overdose, which is indicated by significant respiratory depression [55]. The device also differs in its wear. Rather than being worn, wearable naloxone auto-injectors are placed under the skin, exposing the user to potential infection, tissue injury, and leakage of drug systemically [56]. 

The Automatic Antidote Delivery Device (A2D2) is a proof-of-concept wearable naloxone auto-injector. The subcutaneous device releases the antidote using a portable and wearable magnetic field generator, and trials have shown the drug is released within 10 s [55]. A study conducted in Philadelphia resulted in 76% of participants being in favor of the device and preferring its subtle, comfortable, and discreet design [57,58]. Respondents to this survey also indicated the importance of making such devices available to people experiencing homelessness [57].

Wearable technologies, limited by the current stages of development they are in, also face scrutiny due to viability and stigma. While designed to be discreet, these technologies are to be worn daily by the individual (Table 1), and some people who use drugs will not be comfortable being seen wearing this device or learning to maintain its components. Additionally, the prices of these technologies may be out of range for many individuals if they are expected to pay for them. 

## 4. Discussion

ODTs are emerging solutions to the overdose crisis and have been successful in reversing overdoses and preventing deaths. Existing ODTs complement ongoing efforts to implement and scale up community-based naloxone programs and supervised consumption spaces. Given the magnitude of solitary drug use in the U.S. and the record number of fentanyl and stimulant-involved overdose deaths that occurred during the COVID-19 pandemic [3], the development and evaluation of these tools are urgently required.

There are three categories of ODTs with varying invasiveness, practicality, and privacy protections. Fixed-location devices work as a form of communication or signaling between the person using drugs and supporting staff to guarantee the person’s privacy but provide assistance if necessary. These anchored devices appear best for restrooms and housing facilities where overdoses may occur, as they leverage existing resources in the built environment (i.e., staff) to respond to potential overdoses without disrupting organizational workflow. Smartphone applications and hotlines are a direct line between an individual who uses drugs and people who are willing and able to respond as quickly as possible. As long as the person who is using drugs has their smartphone nearby, this system will alert the correct people to mobilize assistance to the exact location of a potential overdose, whether that be at home, work, or in public, immediately. In addition to preventing fatal overdoses in people who use drugs in workplaces and homes, reverse motion sensor technologies can also address the concerns of business owners in areas with high rates of drug use, some of whom have closed restrooms to avoid liability if patrons overdose inside [59].

Finally, wearable auto-injector technologies are more invasive but bring an added layer of safety, as truly no one needs to be around for an opioid overdose to be reversed, as naloxone is administered directly from the device to the individual wearing the technology. This final type of technology requires a bit more of a commitment on the part of the individual who is using drugs and costs if scaled up, but it offers flexibility in that no matter where they are, the potential for a fatal overdose is reduced, provided that there are trained bystanders nearby who can respond in the event of a non-opioid emergency.

The limitations of this literature review stem from the dependency on published research, media, and news articles matching the search strategy disclosed in the Materials and Methods section. In this narrative review, we restricted our search to PubMed, which may not have identified all relevant peer-reviewed literature related to ODTs. Future studies could leverage multiple databases to identify more studies including non-English scientific and unpublished grey literature to support these findings.

## 5. Conclusions

ODTs are a promising set of interventions that could improve community safety and reduce solitary overdose deaths. Each technology identified enables a rapid response to a potential overdose event, detected through different methods. Further research is needed to understand the effectiveness, feasibility, fidelity, and cost-effectiveness of implementing ODTs as a global overdose risk reduction strategy. Only a small number of ODTs are currently commercially available, but many are in development. Organizations could opt for a set of ODTs that fit their needs, taking into consideration new research, pricing, installation, employer requirements, and limitations. These forms of virtual supervision could complement physical supervised consumption spaces and other approaches to overdose prevention. Research, development, and scale-up of the most practical, safe, and cost-effective ODTs is a public health imperative.

## Figures and Tables

**Table 1 ijerph-20-01230-t001:** Overdose detection technologies appropriate for people who engage in solitary drug use.

Type of Overdose Detection Technology	Name	Implementation in the US	Cost for Organizations	Year of Release	Cost for Individuals	Can Alert Bystanders	Can Alert EMS	Additional Notes
**Fixed-** **Location Devices**	The Brave Sensor	CA, OH	$500/year	2018	$0	x	x	Requires installation
	The Brave Button	CA, OH	$100/year	2018	$0	x	x	Requires installation
	South End Clinic Anti- Motion Detector	MA	NA	2018	$0	x		Requires installation and not commercially available
	Push-Activated Intercom Systems	Nationwide	~$400	Various	$0	x	x	Requires installation
**Smartphone Applications and Hotlines**	The Never Use Alone (NUA) Hotline	ME, MA, NH, RI, VT	$0	2019	$0		x	Requires a charged phone
	The National Overdose Response Service (NORS)	NA (Canada)	$0	2020	$0		x	Requires a charged phone and not available in the US
	The Canary—Prevent Overdose App	Nationwide	$0	2018	$0	x	x	Requires a charged smartphone
	The Brave App	Nationwide	$0	2020	$0	x	x	Requires a charged smartphone
	UnityPhilly	Philadelphia	$0	2019	$0	x	x	Requires a charged phone and not commercially available
	Lifeguard App	NA (British Columbia)	$0	2021	$0	x	x	Requires a charged smartphone and not commercially available in the US
	Second Chance App	Nationwide	$0	2019	$0	x	x	Requires a charged smartphone
	iKeepr	Nationwide	$0	2022	$0	x	x	Requires a charged smartphone
**Wearable Technology**	Wearable Biosensors	Nationwide	$0	Various	~$1000–5000			Requires individuals to wear the device
	Wearable Naloxone Auto Injectors	Nationwide	$0	Various	~$100–200			Requires individuals to wear the device and not commercially available

## Data Availability

Not applicable.

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
