# Peer review of "Overdose Detection Technologies to Reduce Solitary Overdose Deaths: A Literature Review"

_ijerph, 2023, doi:10.3390/ijerph20021230_

Round 1

Reviewer 1 Report

Authors provide a review of overdose detection technologies to reduce overdose among people who are using drugs alone.  This topic for a review is very timely and important as overdose deaths continue to surge and access to drug consumption spaces are limited. 

Some major considerations:

1)    Clarify whether this is a systematic or narrative review and adhere to published guidance in structuring, formatting and describing the methods of  such a paper. See comment and links below.

2)    Consider broadening the search to include more papers in the review as suggested below

3)    The following recently published review article: Matskiv G, Marshall T, Krieg O, Viste D, Ghosh SM. Virtual overdose monitoring services: a novel adjunctive harm reduction approach for addressing the overdose crisis. CMAJ. 2022 Nov 28;194(46):E1568-E1572. doi: 10.1503/cmaj.220579. PMID: 36442886.

Should be incorporated into the introduction. 

How does this review article build on and add to this just-published review article?

Specific considerations and suggestions:

Abstract – The first two sentences construction with the verb “remain” refers to a time window which is not anchored on either end. Be more specific.

Introduction – Again, consider a more specific term and sentence construction than one built around the verb “remain.”

One thing that may be missing from this introduction is acknowledging that many of these technologies have been available for more than four years, have proven that they can save lives, but have not taken off/gone viral as ways to keep people safe. More research is warranted to understand why people do not engage with the existing technologies.

2nd paragraph, first sentence – Instead of “other opioids,” I think it would be clearer if you stated “opioid agonists.”

2nd paragraph, second sentence – Be more specific than “naloxone is effective with opioid misuse” – Naloxone reverses opioid overdoses, but does not reverse the effects of benzodiazepines, cocaine or amphetamines.

2nd paragraph, third sentence – Instead of saying xylazine is linked to overdose deaths (I do not think it has been clearly linked to overdose deaths), it would be more accurate to say that xylazine is frequently combined with opioids, especially fentanyl, and can contribute to sedation when people experience opioid overdose.  This sedation from xylazine will not be reversed by naloxone.

3rd paragraph, 2nd sentence – Not all supervised consumption sites have medical supervision – some are supervised by non-medical peers who are trained in overdose response. I would remove the word “medical,”

3rd paragraph, 3rd sentence – ODTs is an acronym that has not been defined. Please indicate what it stands for – I assume it is overdose detection technologies, but it should be clear for the reader.

Methods – Other key words to consider searching for: “Virtual monitoring” in Pubmed returns: Matskiv G, Marshall T, Krieg O, Viste D, Ghosh SM. Virtual overdose monitoring services: a novel adjunctive harm reduction approach for addressing the overdose crisis. CMAJ. 2022 Nov 28;194(46):E1568-E1572. doi: 10.1503/cmaj.220579. PMID: 36442886.

“Virtual spotting” – in Google Scholar it returns - Matskiv G, Marshall T, Krieg O, Viste D, Ghosh SM. Virtual overdose monitoring services: a novel adjunctive harm reduction approach for addressing the overdose crisis. CMAJ. 2022 Nov 28;194(46):E1568-72.

“Drug spotting” in pubmed returns: Perri M, Kaminski N, Bonn M, Kolla G, Guta A, Bayoumi AM, Challacombe L, Gagnon M, Touesnard N, McDougall P, Strike C. A qualitative study on overdose response in the era of COVID-19 and beyond: how to spot someone so they never have to use alone. Harm Reduct J. 2021 Aug 5;18(1):85. doi: 10.1186/s12954-021-00530-3. PMID: 34353323; PMCID: PMC8339679.

“Reverse Motion Detector” – which returned in Pubmed: Schreyer KE, Malik S, Blome A, D'Orazio JL. A Case Report of a Novel Harm Reduction Intervention Used to Detect Opioid Overdose in the Emergency Department. Clin Pract Cases Emerg Med. 2020 Nov;4(4):548-550. doi: 10.5811/cpcem.2020.7.47936. PMID: 33217269; PMCID: PMC7676791 which cited: Fozouni L, Buchheit B, Walley AY, Testa M, Chatterjee A. Public restrooms and the opioid epidemic. Subst Abus. 2020;41(4):432-436. doi: 10.1080/08897077.2019.1640834. Epub 2019 Aug 1. PMID: 31368865.

“bathroom” and “Overdose” in pubmed returns these:

Buchheit BM, Crable EL, Lipson SK, Drainoni ML, Walley AY. "Opening the door to somebody who has a chance." - The experiences and perceptions of public safety personnel towards a public restroom overdose prevention alarm system. Int J Drug Policy. 2021 Feb;88:103038. doi: 10.1016/j.drugpo.2020.103038. Epub 2020 Nov 21. PMID: 33232885.

van Draanen J, Satti S, Morgan J, Gaudette L, Knight R, Ti L. Using passive surveillance technology for overdose prevention: Key ethical and implementation issues. Drug Alcohol Rev. 2022 Feb;41(2):406-409. doi: 10.1111/dar.13373. Epub 2021 Aug 5. PMID: 34355446.

Wolfson-Stofko B, Gwadz MV, Elliott L, Bennett AS, Curtis R. "Feeling confident and equipped": Evaluating the acceptability and efficacy of an overdose response and naloxone administration intervention to service industry employees in New York City. Drug Alcohol Depend. 2018 Nov 1;192:362-370. doi: 10.1016/j.drugalcdep.2018.08.001. Epub 2018 Sep 21. PMID: 30287108; PMCID: PMC6237076.

Wolfson-Stofko B, Bennett AS, Elliott L, Curtis R. Drug use in business bathrooms: An exploratory study of manager encounters in New York City. Int J Drug Policy. 2017 Jan;39:69-77. doi: 10.1016/j.drugpo.2016.08.014. Epub 2016 Oct 18. PMID: 27768996; PMCID: PMC5304450.

Searching google scholar for “bathroom overdose” gave these papers: Wolfson-Stofko B, Bennett AS, Elliott L, Curtis R. Drug use in business bathrooms: An exploratory study of manager encounters in New York City. Int J Drug Policy. 2017 Jan;39:69-77. doi: 10.1016/j.drugpo.2016.08.014. Epub 2016 Oct 18. PMID: 27768996; PMCID: PMC5304450.

Gaeta JM. A Pitiful Sanctuary. JAMA. 2019 Jun 25;321(24):2407-8.

Wolfson-Stofko B, Curtis R, Fuentes F, Manchess E, Rio-Cumba D, Bennett AS. The portapotty experiment: neoliberal approaches to the intertwined epidemics of opioid-related overdose and HIV/HCV, and why we need cultural anthropologists in the South Bronx. Dialectical anthropology. 2016 Dec;40(4):395-410.

Missing from the Methods is an explanation of the format for this review. Is it a “systematic review”? If so it would improve the paper if the methods were guided by the PRISMA 2020 guidance on systematic review: https://systematicreviewsjournal.biomedcentral.com/articles/10.1186/s13643-021-01626-4

If it is a narrative review, then here is a scale the authors could use to structure the methods: https://researchintegrityjournal.biomedcentral.com/articles/10.1186/s41073-019-0064-8

Results:

3.1 Fixed Location devices Requiring Physical Installation

I believe the South End Clinic motion detector timer is initiated by the closing of the door. “These types of sensors remove the responsibility of overdose surveillance from the organization” – I think that this should read “These types of sensors enhance the ability of organizations to provide higher quality overdose surveillance”

Since there isn’t a limitations section in this paper, I would suggest raising potential limitations within each section. In this case, the Brave button requires the PWUD to plan to hit the button before doing their shot, which could be happening 10-18 times/day. It is an extra step that would need training and reinforcement by the PWUD.

Lines 159-160: I would say that they enter to check on the wellbeing of the participant and are able to administer naloxone in the case of an overdose.

3.2 Mobile Applications and Hotlines

A physical device is still necessary in these cases – people need to have a CHARGED phone (regular flip phone in case of NUA or a smart phone in case of Brave App or Canary) and this is a barrier

This history of NUA needs to be edited for accuracy. The NUA hotline originated as a national concept in August 2019. It was advertised from the beginning as being available in all 50 states. Over the course of the next year, as COVID happened and volume increased, State hotlines began springing up. There was a NUA New York hotline (started by this reviewer -SM) and also a Never Use Alone Massachusetts and Never Use Alone Vermont. These all started in April 2020. In Mid-2021, Never Use Alone Massachusetts and Never Use Alone Vermont merged into NUA New England. Anyone from any state can contact any of these hotlines and receive the same service, though the phone numbers are advertised by state. Many of the operators work on multiple hotlines. One major piece of information missing in this section is that NUA uses a toll-free 1-800 number, that does not require the caller to have a smart phone or have minutes on their phone to make a call.  Overall, there have been something like 14,000 calls with over 100 overdoses across all of the hotlines, but I would contact Never Use Alone via Facebook to get their most recent stats.

“The expansion from Massachusetts to the broader New England area is a result of recent exposure and increased volunteers, and the hotline looks to expand coverage to the entire U.S.” This is incorrect. They have received calls from all over the country, and a heat map was published back in 2021 showing callers.  (https://twitter.com/stephenHRNRP/status/1409847494687825924?s=20&t=YFmb5jX9spfHz56DyZ2FAw) One of the major limitations is that the response time by EMS can vary greatly, especially in rural areas, so this is not a 100% failsafe method. It also utilizes a person’s level of responsiveness, not their breathing, as the way to alert someone may be overdosing, and this has resulted in activations of EMS that were probably not needed. This is done out of an abundance of caution, but also can deter PWUD away from utilizing the service.

Canary: “The user places their phone in their pocket once activated. The device detects and tracks small body movements, and if movements consistent with respiration are undetectable, Canary warns the 197” – I know that this is what they advertise, but I would encourage you to test this feature out. I activated canary while laying down, and put the phone directly on my chest. It does not reset the timer, even when I’m breathing regularly. I even put it on my stomach and did dramatic belly breaths and it did not reset the time. The only thing that reset it is when I moved it with my hand. To me, this really is something that is monitoring level of responsiveness, rather than monitoring respiratory effort. One limitation is that it relies on the person have a pre-established contact who is ready to intervene either physically or by calling 911, and if that isn’t confirmed prior to activating, no one will know that the person needs help and there is no back up to call EMS.

For the BraveApp, the text is not clear about whether “first contact” and “supporter” are the same person or different people.  If they are the same, then it is confusing to imagine how EMS would “check in through the call…[and] request access to their game plan …[and] activates the game plan.” Also  regarding“the entire process be kept private and anonymous,” does this mean that the first contact/supporter is going to be aware beforehand of the person’s game plan?  How can they be counted on to activate the game plan, if they have not been told about the plan beforehand? I would reach out to Brave App for more information on Overdose detections. They had at least 150 as of May 2022 (https://fb.watch/hm_LMy1ufI/) but that is already 7 months out of date.

With UnityPhilly, has there been any assessment of premature or false naloxone administrations and rates of precipitated withdrawal? This app does not appear to be currently functional. I downloaded it from the app store and you need a username and password and there is no option to sign up from the app.

Lifeguard appears to work similarly to Canary, maybe these should be grouped. It may be worth splitting these up between “Third Party Monitored ODTs including NUA/Brave/Unity” and “User Defined Responder ODTs” or something like that. Is this only available in Canada? I can’t find it in the Apple App store.

For the Second Chance App that uses sound waves, can you speak to the advantages and differences reported between using the phones accelerometer in Canary vs. sound waves in Second Chance for motionless detection?  I could not find this in the Apple App store.

“The inequities of smartphone access among people who use drugs, specifically for those who are unhoused, remain a challenge to widespread adoption” This is actually something that should be labeled as a limitation for all of the App based programs, not just this one.

I had tried to search “IKeeper” on the app store, but it is actually “IKeepr” with no e. I attempted to use the app but it kept freezing every time I put in my phone number.

3.3 Wearable Technology

“In studies years ago” is too vague – please state specifically when you are referring to.

Provide some more rationale or context for this statement “In comparison to smartphones, personal data assistants, and beepers, wearable ODTs were noted as potentially stigmatizing electronic devices [50].”  It seems like wearables could be incorporated into existing watches or exercise monitors or be worn discretely underneath clothing.  It is not clear to me what would be stigmatizing to these devices.

Paragraph 3: I’m not sure I understand the context being given here about these devices. Who is this information streaming to? How does it help if information streams to a smartphone if there is no mechanism to activate help? How does a device that measures opioid use (perhaps for a doctor?) help to respond to overdose? The way you describe it as “tracking trigger points for opioid use, relapse, or overdose directly on the individual” sounds more like surveillance by medical providers rather than a harm reduction tool.

Embrace: “streams information to the user’s smartphone to accurately indicate the user’s moments of greatest need” – what does that mean?

For the naloxone auto-injectors, it seems like the reference point for these devices would be the insulin pump, which administers insulin subcutaneous based on blood sugar readings without substantial risk of infection and attached to the skin in discrete locations.

Discussion

In the Results, the only approach for which any numbers or rescues were presented was for Never Use Alone. Have all of the technologies presented “been successful in reversing overdoses and preventing deaths”?  I think it would be worthwhile making clear which have and which not at this point in time, which are actually in the field and there are reports of them working, and which are still in development.  For which there are data and for which there are not data and the quality of these data.

Last paragraph: instead of “these technologies,” specify the reverse motion sensor technology. Saying “these technologies” sounds like you’re including the apps in creating peace of mind for businesses.

How does xylazine impact the effectiveness of apps that rely on motion/patient response rather than observed work of breathing or pulse oximetry? Most people who are using xylazine will experience extreme sedation that may not be accompanied by respiratory depression or arrest – how can we adapt these technologies to account for this to reduce false activations of responders and EMS?

Conclusion:

Given that more than half of these don’t appear to be currently operational, I’m not sure you can conclude that “a variety of ODTs exist” – there are pretty limited, functional options still.

Reviewer 2 Report

This was a nicely written literature review on technology to reduce injury and death by overdose. I have no doubt this is an area where research and development investment are needed. Well done

I do have couple of questions/ comments though:

Q1: One of the issues I had with this is the fact that only PubMed database was used to search for studies. Any reason why this is the case? I understand this is a literature review and not a higher order review, however, I wonder if including other databases may have added value to the overall findings.

Q2: in your conclusion you almost start by addressing limitations of the study "The limitations of this literature review stem from the dependency on published research, media, and news articles matching the search strategy disclosed in the Materials and Methods section" but really this should be towards end of your discussion and not in the conclusion or at least not so early in the conclusion. I suggest the authors address the above limitation towards end of the discussion and add to that the fact they have only searched one database may have impacted the validity of the findings as some studies may have been missed. another limitation is that being a literature review the evidence presented here may not be reliable although it does warrant further work including more systematic search and analysis etc. 

I think your conclusion should continue to summarize the findings as you have done without the limitation section, which is way too early in the segment. 
